# Quantitative Assessment of CD200 and CD200R Expression in Lung Cancer

**DOI:** 10.3390/cancers13051024

**Published:** 2021-03-01

**Authors:** Ioannis A. Vathiotis, Tyler MacNeil, Jon Zugazagoitia, Konstantinos N. Syrigos, Thazin Nwe Aung, Aaron M. Gruver, Peter Vaillancourt, Ina Hughes, Steve Hinton, Kyla Driscoll, David L. Rimm

**Affiliations:** 1Department of Pathology, BML 116, Yale University School of Medicine, 310 Cedar St. P.O. Box 208023, New Haven, CT 06520-8023, USA; ioannis.vathiotis@yale.edu (I.A.V.); tyler.macneil@yale.edu (T.M.); jon.zugazagoitia@yale.edu (J.Z.); thazin.aung@yale.edu (T.N.A.); 2Department of Medicine, National and Kapodistrian University of Athens School of Medicine, 11527 Athens, Greece; ksyrigos@med.uoa.gr; 3Eli Lilly and Company, Indianapolis, IN 46285, USA; gruver_aaron_m@lilly.com (A.M.G.); vaillancourt_peter@lilly.com (P.V.); hughes_ina_e@lilly.com (I.H.); steve.hinton@lilly.com (S.H.); driscoll_kyla@lilly.com (K.D.)

**Keywords:** CD200, CD200R, NSCLC, lung cancer, immunotherapy

## Abstract

**Simple Summary:**

Most patients with advanced non-small cell lung cancer (NSCLC) do not respond to programmed cell death protein 1 (PD-1)/programmed death-ligand 1 (PD-L1) axis blockade, highlighting the need for the identification of new targets for immune checkpoint inhibition. In this study, we used quantitative immunofluorescence to characterize the CD200/CD200R immune checkpoint in lung cancer. We described a careful validation of antibody probes for this pair and found CD200 and CD200R to be overexpressed in 29.7% and 25% of NSCLC patients, respectively; stromal expression of CD200R was significantly increased in patients with squamous histology. Additionally, we showed that PD-L1 was moderately correlated with CD200 but only weakly correlated with CD200R. As new drugs targeting the CD200/CD200R interaction enter into clinical trials in humans, this work identifies this immune checkpoint as a potential target for patients with NSCLC and lays the groundwork for the development of a clinical companion diagnostic test.

**Abstract:**

CD200/CD200R is an immune checkpoint with broad expression patterns and a potential target for immune therapy. In this study, we assess both CD200 and CD200R expression in solid tumors, with a focus on lung cancer, and evaluate their association with clinicopathologic characteristics, mutation status, outcome, and programmed death-ligand 1 (PD-L1) expression. We used multiplexed quantitative immunofluorescence (QIF) to measure the expression of CD200 and CD200R in a total of 455 patients from three lung cancer cohorts. Using carefully validated antibodies, we performed target measurement with tyramide-based QIF panels and analyzed the data using the PM2000 microscope and AQUA software. CD200 tumor positivity was found in 29.7% of non-small cell lung cancer (NSCLC) patients and 33.3% of lung large cell neuroendocrine carcinoma (LCNEC) patients. CD200 demonstrated notable intratumoral heterogeneity. CD200R was expressed in immune cells in 25% of NSCLC and 41.3% of LCNEC patients. While CD200R is predominantly expressed in immune cells, rare tumor cell staining was seen in a highly heterogeneous pattern. CD200R expression in the stromal compartment was significantly higher in patients with squamous differentiation (*p* < 0.0001). Neither CD200 nor CD200R were associated with other clinicopathologic characteristics or mutation status. Both biomarkers were not prognostic for disease-free or overall survival in NSCLC. CD200 showed moderate correlation with PD-L1. CD200/CD200R pathway is frequently expressed in lung cancer patients. Differential expression patterns of CD200 and CD200R with PD-L1 suggest a potential role for targeting this pathway alone in patients with NSCLC.

## 1. Introduction

Immune checkpoint blockade has become the new standard in the treatment of advanced-stage non-small cell lung cancer (NSCLC). Nevertheless, programmed cell death protein 1 (PD-1)/ programmed death-ligand 1 (PD-L1) axis inhibitor monotherapy only benefits a minority (29.6%) of selected NSCLC patients in the long term [1]. Even when combined with standard platinum-based chemotherapy, only 47.6% of NSCLC patients respond to PD-1/PD-L1 axis inhibitors [2]. These profound and durable responses, yet limited to a restricted number of patients, have underlined the need of identifying new potential targets for immune checkpoint inhibition, which could potentially be targeted either alone or in combination with the PD-1/PD-L1 pathway, expanding patient populations that can benefit from immunotherapy.

CD200 is a member of the immunoglobulin superfamily that is expressed on various cell types including tissue stem cells, thymocytes, neurons, hair follicular cells, immune cells, endothelial cells, and cancer cells [3,4,5]. CD200 receptor (CD200R) is mainly found on immune cells of monocyte-macrophage lineage such as monocytes, macrophages, dendritic cells and basophils; lower levels of CD200R expression have also been reported on T, B and NK cells [6,7,8]. CD200/CD200R interaction functions as an immune checkpoint with broad expression patterns. It ultimately leads to the attenuation of inflammation with implications to autoimmunity and preservation of self-tolerance, suppression of immune response following infections or allergic reactions, graft acceptance post transplantation, and tumor cell mediated immune suppression [9]. Interestingly, intracellular CD200R signaling is mediated by Dok1 and Dok2 and differs from other previously described checkpoints including PD-1 and cytotoxic T-lymphocyte-associated protein 4 (CTLA-4) that employ immunoreceptor tyrosine-based switch motifs (ITSMs). In addition, while PD-1/PD-L1 pathway is mostly dependent on modulating T cell activity directly and inducing IFNγ, CD200/CD200R pathway modulates immune activity largely via myeloid suppressive mechanisms and is not dependent solely on IFNγ activity [10,11,12].

Both CD200 and CD200R have been thoroughly studied in hematologic malignancies; CD200 is an independent negative prognostic factor for patients with chronic lymphocytic leukemia (CLL), acute myeloid leukemia (AML), multiple myeloma (MM) and myelodysplastic syndromes (MDS) [13,14,15,16]. This pathway has also been characterized in some solid tumors, such as melanoma and neuroblastoma [17,18,19]. Here, we used multiplexed quantitative immunofluorescence (QIF) to assess the distribution and patterns of expression of both CD200 and CD200R in three independent lung cancer cohorts and evaluate their association with clinicopathologic characteristics, mutation status, outcome, and PD-L1 expression.

## 2. Results

For the anti-CD200R antibody validation process, we used two different antibodies targeting non-overlapping epitopes of the extracellular domain of human CD200R, Lilly antibody 1 and Lilly antibody 2 (Eli Lilly and Company) (Appendix A). First, we titrated these antibodies in a NSCLC array (YTMA295) containing 35 tumor cores with presumed variable CD200R expression (Appendix A). We observed CD200R staining within the stromal compartment, with membranous localization of signal (Appendix A). We also compared the CD200R QIF scores obtained with both antibodies on YTMA295 showing a high correlation coefficient (R^2^ = 0.80) (Appendix A). Then, we built a custom CD200R index tissue microarray (TMA; YTMA424) with 0.6 mm cores for assay validation and reproducibility assessments. YTMA424 contained both cancerous (lung, pancreatic, breast, ovarian cancer, and lymphoma) and unmatched normal tissue as well as formalin-fixed paraffin-embedded (FFPE) prepared CHO cells (CHO parental) and a CHO cell line which overexpressed CD200R (CHO MO2O-A5) (Appendix A). Cell line TMA construction has been published in detail elsewhere [20]. We used this TMA to recalculate the signal/noise ratio for the optimal concentration of our antibodies, and this data suggested that Lilly antibody 1 had better signal/noise compared to Lilly antibody 2 (Appendix A) [21]. In addition, we confirmed good reproducibility between the two antibodies (R^2^ = 0.92) and two independent experiments with Lilly antibody 1 (R^2^ = 0.82) (Appendix A). As a final validation step, we stained a multi-tumor TMA (YTMA395) containing 295 cores from 12 different types of solid tumors and hematologic malignancies to ensure the reproducibility of our antibodies across different tumor types (R^2^ = 0.59) (Appendix A). Thus, Lilly antibody 1 was considered validated and was used for the remainder of our experiments. We followed a similar process to validate the anti-CD200 antibody that we used in our experiments (clone 333), which is summarized in Appendix A.

Apart from a useful validation step, staining YTMA395 also served as a screening step to identify human solid tumors with increased CD200 and CD200R expression levels. Although stromal CD200R expression appeared uniform, as expected, across different tumor types, we measured elevated levels of CD200 within the tumor compartment in patients with lung and pancreatic cancer (Appendix A). Therefore, we decided to focus our efforts to characterize this pathway on lung cancer. QIF scores of CD200 and CD200R showed a continuous distribution in both the tumor and stromal compartments and were comparable between all three lung cancer cohorts. We observed a predominantly membranous or cytoplasmic CD200 staining pattern in both tumor and stromal cells (Figure 1C–E). Subsequently, we employed a visual cutpoint (defined as the lowest expressing case where the signal is discernably present and specific above the background noise) to define CD200 tumor positivity; 29.7% of NSCLC cases and 33.3% of lung large cell neuroendocrine carcinoma (LCNEC) cases were found positive for CD200 within the tumor compartment (Figure 1A,B). CD200 expression exhibited notable intratumoral heterogeneity (R^2^ = 0.09–0.31 between different blocks of YTMA423 and R^2^ = 0.51 between serial sections of the same block of YTMA423) (Figure 1F–H). In the stroma, most of the cases were positive for CD200, to some degree. A good correlation between tumor and stromal expression of CD200 (R^2^ = 0.69) was noted as well (Figure 1I). While CD200R was predominantly expressed on the membrane of stromal cells, rare membranous staining of tumor cells was seen in a highly heterogeneous pattern (Figure 2C–E). Visually, 25% of NSCLC cases and 41.3% of LCNEC cases were positive for CD200R in the stroma (Figure 2A,B). CD200R expression within the stromal compartment was quite consistent (R^2^ = 0.46–0.60 between different blocks of YTMA423 and R^2^ = 0.94 between serial sections of the same block of YTMA423) (Figure 2F–H). The correlation between tumor and stromal expression of CD200R was lower (R^2^ = 0.54), in part because of the scarce and irregular pattern of expression of this biomarker in tumor cells (Figure 2I). The correlation between CD200 and CD200R was weak in general. The correlation coefficient was highest when both biomarkers were measured within the stromal compartment (R^2^ = 0.31) (Appendix A).

Next, we assessed the association of CD200 and CD200R with clinicopathologic characteristics in patients with NSCLC. CD200R expression in the stromal compartment was significantly increased in patients with squamous differentiation (*p* < 0.0001 vs. non-squamous) (Appendix A). No other significant or consistent association between tumor or stromal expression of both markers and disease stage or smoking status was noted (Appendix A). Neither tumor nor stromal expression of CD200 or CD200R demonstrated significant associations with epidermal growth factor receptor (*EGFR*) or *KRAS* mutation status in NSCLC patients (Appendix A).

Based on the data on hematologic malignancies, we then examined the prognostic significance of this immune checkpoint in NSCLC. We dichotomized the continuous QIF scores of CD200 and CD200R into high and low statuses using the median as a cutoff. Tumor CD200 expression did not significantly predict disease-free survival (DFS) (*p* = 0.84) after surgery (Figure 3A) or overall survival (OS) (*p* = 0.55) (Figure 3C) in early-stage NSCLC patients. Similarly, neither did stromal expression of CD200R significantly predict DFS (*p* = 0.33) after surgery (Figure 3B) nor OS (*p* = 0.42) (Figure 3D) in early-stage NSCLC patients.

Because of the different types of immune cells that they primarily affect, as well as their distinct intracellular mechanisms of action, CD200/CD200R and PD-1/PD-L1 immune checkpoints could potentially be targeted either synergistically or independently. To evaluate the association between CD200, CD200R, and PD-L1 in NSCLC, we stained serial sections of YTMA423 for all three biomarkers (Figure 4A–C). Visually, we found evidence of colocalization of all three markers in both tumor cells and immune cells in the stroma. PD-L1 expression showed weak correlation with CD200 expression when both were measured within the tumor compartment (R^2^ = 0.28) and moderate correlation when measured within the stromal compartment (R^2^ = 0.40). We noticed a number of cases that were positive for PD-L1 but negative for CD200. This phenomenon was more prominent within the tumor than the stromal compartment (Figure 4D,E). On the contrary, CD200R in stroma did not correlate with PD-L1 in tumor (R^2^ = 0.07) or stroma (R^2^ = 0.10). In that case, the trend was bimodal with cases that were positive for one marker but negative for the other and vice versa (Figure 4F,G).

## 3. Discussion

The CD200/CD200R ligand receptor pair is a provocative immune-regulatory target, but its protein expression pattern in solid tumors is not well established. In this study, we used QIF to characterize the CD200/CD200R expression with a focus on lung cancer. We developed and validated two different immunofluorescence panels with primary antibodies to detect epithelial tumor cells and either CD200 or CD200R on the same tissue section. We showed that CD200 and CD200R are expressed on both tumor and stromal cells of NSCLC patients. CD200 exhibited a rather heterogeneous pattern of expression, with areas staining positive and others staining negative even within the same tumor. Stromal CD200R staining was consistent among different cores of the same tumor. We observed significantly increased infiltration of CD200R-positive immune cells in the stroma of NSCLC patients with squamous tumors, in comparison with those with nonsquamous tumors. Other than that, we found no clear or consistent association between CD200 or CD200R expression and clinicopathologic characteristics, genomic features or outcome. We also described the weak correlation between these two biomarkers in NSCLC.

Recently, Yoshimura et al. explored the clinicopathologic and prognostic implications of the CD200/CD200R immune checkpoint in 632 NSCLC patients by immunohistochemistry (IHC) [22]. In this study, CD200 expression was rarely seen in the stroma, with 93.3% of cases showing absence of stromal CD200 staining. Increased expression of CD200 in tumor was associated with female sex, never-smoking status, adenocarcinoma histology, early disease stage, and *EGFR* mutations. On the contrary, increased expression of CD200R in both tumor and stroma was associated with male sex, ever-smoking status, non-adenocarcinoma histology, and advanced disease stage, while low CD200R expression was associated with *EGFR* mutations. Furthermore, high CD200 and low CD200R expression was associated with better survival outcomes. However, the above findings were not validated by our study. Overall, the discordances between the two studies can be explained by a number of reasons. First, we only used carefully validated monoclonal antibodies and performed assay optimization with regards to primary antibody concentration. Furthermore, each of the 455 lung cancer patients included in our study was represented in threefold redundancy; QIF scores for each patient were averaged and then correlated with clinicopathologic characteristics and outcome. This is particularly important not only in the context of TMA studies, but also for biomarkers that exhibit considerable heterogeneity, as is the case for CD200. Last but not least, we used continuous QIF scores, while Yoshimura et al. used a semi-quantitative method to categorize staining intensity, which may be subject to inter-rater variability.

Apart from NSCLC, we examined the expression of both biomarkers in patients with lung LCNEC. Even though CD200 is expressed in neuroendocrine tissues, we only detected a slight increase in the percentage of LCNEC cases staining positive for CD200, in comparison with NSCLC cases (33.3% for LCNEC versus 29.7% for NCSLC). This comes in contrast to a previous study from Love et al. that supported that CD200 may act as a marker of neuroendocrine differentiation and tends to be overexpressed in lung among other neuroendocrine neoplasms. That study reported that the percentage of CD200 positive neuroendocrine tumors can be as high as 87%, but was underpowered for lung LCNEC tumors [23].

While stromal cells express both CD200 and CD200R, future work is underway to more specifically define the immune cell populations that express each protein. CD200R is principally expressed on immune cells of monocyte-macrophage lineage, with an emphasis on myeloid-derived suppressor cells (MDSCs) [19,24]. Additionally, Gaiser et al. found evidence of CD200R-expressing macrophages in Merkel cell carcinoma and suggested a role for the induction of an immunosuppressive M2 phenotype [25]. This is supported, diagonally, by the fact that we detected immune cells staining double positive for both CD200R and PD-L1. Similarly, work from our laboratory showed CD68-positive macrophages as the predominant immune cell type that expresses PD-L1 in NSCLC [26]. Furthermore, we have produced preliminary data indicating that CD200R colocalizes with both CD68 and CD163 in the stroma and further work is currently in progress to validate and examine in detail these findings.

The CD200/CD200R pathway has been very well studied in hematologic malignancies, where CD200 can assist in diagnosis and staging of individuals with CLL [27]. CD200 also represents a marker of disease progression and has been identified as a negative prognostic factor for patients with CLL, AML, MDS, and MM [13,14,15,16]. Accumulating data have begun to shed light on the importance of CD200/CD200R signaling in patients with solid tumors. This pathway is potentially involved in immunological quiescence of the central nervous system. CD200 protein was found to be expressed on a variety of human brain tumors; serum CD200 concentration levels were highest for patients with glioblastoma multiforme and correlated with the expansion of MDSCs. Moreover, CD200 was overexpressed in pediatric neuroblastoma tumors. Tumor CD200 expression was correlated with significantly decreased tumor infiltration from CD4-positive and CD8-positive T cells and decreased production of IFNγ and TNFα [19]. CD200R antagonism inhibited the expansion of MDSC, impeded tumor growth and significantly prolonged survival in tumor bearing mice [28]. CD200R deficiency also resulted in expedited growth of CD200-positive B16 melanoma tumors in mice; such tumors exhibited significantly decreased infiltration of CD4-positive and CD8-positive T cells with concomitant expansion of MDSCs and FOXP3-positive regulatory T cells and reduction of NK cells in the liver [18].

Recently published preclinical data illustrated that inhibition of the CD200/CD200R interaction can induce chemokine response, stimulate dendritic cell differentiation, enhance antigen-specific response and ultimately lead to downregulation of PD-1 receptors in glioblastoma multiforme cells [29]. Additionally, in vivo blockade of this pathway in pancreatic ductal adenocarcinoma animal models significantly reduced intratumoral infiltration and suppressive activity of MDSCs, limited tumor growth, and enhanced the efficacy of PD-1/PD-L1 axis inhibitors [24]. To our knowledge, our study is the first to assess the correlation between CD200/CD200R and PD-1/PD-L1 pathways in human NSCLC. We found that CD200, but not CD200R, was moderately correlated with PD-L1, and this correlation was strongest within the stromal compartment. This is a finding of potential importance, as new drugs, targeting this pathway, enter into phase I clinical trials in patients with hematologic malignancies as well as solid tumors. Samalizumab, which is a first-in-class monoclonal antibody against CD200, showed positive preliminary results in a phase I study in patients with advanced CLL and MM, which warrant its further development as an immune checkpoint inhibitor [30].

There are several limitations in this study. First, this is a single-institutional, retrospective collection. While treatments are similar by stage and represent the standard of care, there is some heterogeneity in treatment since the collection is not derived from a clinical trial. Sample size, while acceptable for NSCLC, is limited for lung neuroendocrine neoplasms, and small-cell carcinomas and pulmonary carcinoids were not represented at all. Additionally, our NSCLC cohort (YTMA423) contained predominantly early-stage (stages I–IIIA) patients that received surgical treatment for their disease, so future studies addressing the distribution and prognostic value of these biomarkers in NSCLC will be required to address advanced-stage NSCLC patients. Finally, in this study we used TMAs rather than whole tissue sections. Therefore, we tried to alleviate the issue of tumor heterogeneity by staining tumor cores in threefold redundancy (each patient was represented by three histospots cored from different tumor regions). This approach is common for initial studies, even though companion diagnostic testing always requires whole tissue sections.

## 4. Materials and Methods

### 4.1. Patient Cohorts and TMA Construction

Retrospectively collected, pretreatment, FFPE tumor specimens represented in TMA format from three independent lung cancer cohorts from Yale were analyzed [31]. Clinical annotations were retrieved from clinical records and pathology reports. Cohort 1 (YTMA423) contained specimens from 287 early-stage NSCLC patients with tumors resected between 2011 and 2016; cohort 2 (YTMA355) contained specimens from 30 patients with LCNEC of the lung with the majority of the tumors resected between 2007 and 2015; cohort 3 (YTMA310) contained specimens from 138 NSCLC patients with known *EGFR* and *KRAS* genotypes with tumors resected between 2011 and 2013 (Appendix A) [32]. YTMA423 was used to assess for biomarkers’ prevalence in NSCLC, correlation with clinicopathologic characteristics, prognostic significance and PD-L1 expression. Table 1 summarizes the baseline characteristics of the patients included in YTMA423. Furthermore, YTMA355 was used to assess for biomarkers’ prevalence in lung LCNEC and YTMA310 to evaluate biomarkers’ association with mutation status. Written informed consent or waiver of consent was provided by all the patients. The study was approved by the Yale Human Investigation Committee protocol #9505008219 and conducted in accordance with the Declaration of Helsinki.

### 4.2. Multiplexed Immunofluorescence Staining Protocol

TMA slides were deparaffinized and subjected to antigen retrieval with ethylenediaminetetraacetic acid buffer (pH 9 and 8 for CD200 and CD200R, respectively) at 97 °C for 20 min in a pressure heating container (PT module, Lab Vision, Fremont, CA, USA). Next, slides were incubated with a solution of 0.3% hydrogen peroxide in methanol for 30 min to inactivate endogenous peroxidase, followed by another 30 min of incubation with 0.3% bovine serum albumin with 0.05% tween-20 blocking solution. Subsequently, a sequential multiplexed immunofluorescence staining was performed with primary antibodies to detect epithelial tumor cells (pan-cytokeratin, polyclonal, Agilent, Santa Clara, CA, USA) and CD200 (clone 333, Sino Biological, Beijing, China), CD200R (Lilly antibody 1, Eli Lilly and Company, Indianapolis, IN, USA) or PD-L1 (clone E1L3N, Cell Signaling Technology, Danvers, MA, USA) on serial TMA sections. Isotype-specific horseradish peroxidase (HRP)-conjugated secondary antibodies and tyramide-based amplification systems were used for signal detection. 4′,6-diamino-2-phenylindole (DAPI) was used to highlight all nuclei. Control slides from index arrays created for this specific purpose were included in each staining experiment to ensure reproducibility.

More details regarding incubation times, antibody clones and concentrations, and fluorescent reagents used can be found in Appendix A.

### 4.3. Fluorescence Signal Quantification

Signal quantification for CD200, CD200R and PD-L1 was determined by the automated quantitative analysis (AQUA™) method of QIF on fluorescence images acquired using a PM-2000 system (Navigate Biopharma, Carlsbad, CA, USA), as previously described [33]. All three targets were measured within two different compartments: (1) the tumor mask, created by binarizing and then dilating the cytokeratin signal; and (2) the stromal mask, created by excluding the tumor mask from DAPI mask, which was also created by binarizing and then dilating the DAPI signal, which represented the total tissue. QIF scores were generated by dividing the summed pixel intensities for the marker of interest by the area of the compartment in which it was measured and then normalized to the exposure time and bit depth at which the images were captured. Cases with staining artifacts or less than 2% compartment area were systematically excluded after visual inspection. For each cohort, staining and target measurement was performed in three independent TMA blocks, with each block containing one nonadjacent tumor core per patient; QIF scores were then averaged for each case.

### 4.4. Statistical Analysis

Pearson correlation coefficient was used to analyze the linear association between two continuous variables. The *t* test or one-way analysis of variance was used to compare the means between two or more groups, respectively. The chi-square test was used to compare proportions. For the survival analysis, tumors were split into high and low according to CD200 and CD200R expression using the median as a cutoff. Survival curves were estimated with the Kaplan–Meier product-limit method and compared by using the log-rank test. Hazard ratios for OS were calculated by using the Cox regression model. All hypothesis testing was performed at a two-sided significance level of α equal to 0.05. Statistical analysis was performed using GraphPad Prism 8 software (GraphPad Software, La Jolla, CA, USA).

## 5. Conclusions

In conclusion, this work demonstrates the distribution and patterns of expression of the CD200/CD200R immune checkpoint in lung cancer as well as its correlation with PD-L1. The multiplexed immunofluorescence assays that we developed for both targets have the potential to evolve into companion diagnostic tests that could be used in the clinic to select patients that will benefit from inhibition of this pathway.

## Figures and Tables

**Figure 1 cancers-13-01024-f001:**
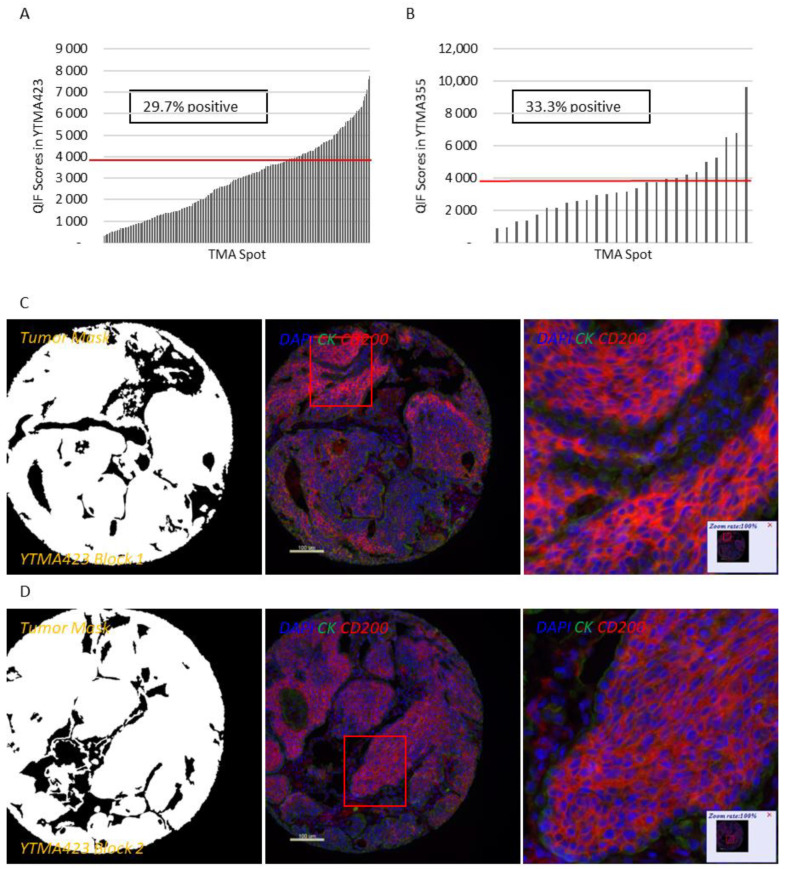
Patterns of expression and distribution of CD200 in lung cancer. (**A**,**B**) Dynamic range charts that depict CD200 expression within the tumor compartment in non-small cell lung cancer (NSCLC) (**A**) and large cell neuroendocrine carcinoma (LCNEC) (**B**) patients. Red lines represent the visual cutoffs that were used to define CD200 positive cases. (**C**–**E**) Different cores from the same representative NSCLC case, stained in YTMA423 blocks 1 (**C**), 2 (**D**) and 3 (**E**). (**F**–**I**) Scatter plots indicative of intratumoral heterogeneity of CD200 in NSCLC. Scatter plots of YTMA423 block 1 versus block 2 (**F**) and block 3 (**G**). Scatter plot between serial cuts of YTMA423 block 1 (**H**). Scatter plot between tumor and stromal expression of CD200 on the same cut (**I**). Abbreviations; DAPI, 4′,6-diamino-2-phenylindole; CK, Cytokeratin.

**Figure 2 cancers-13-01024-f002:**
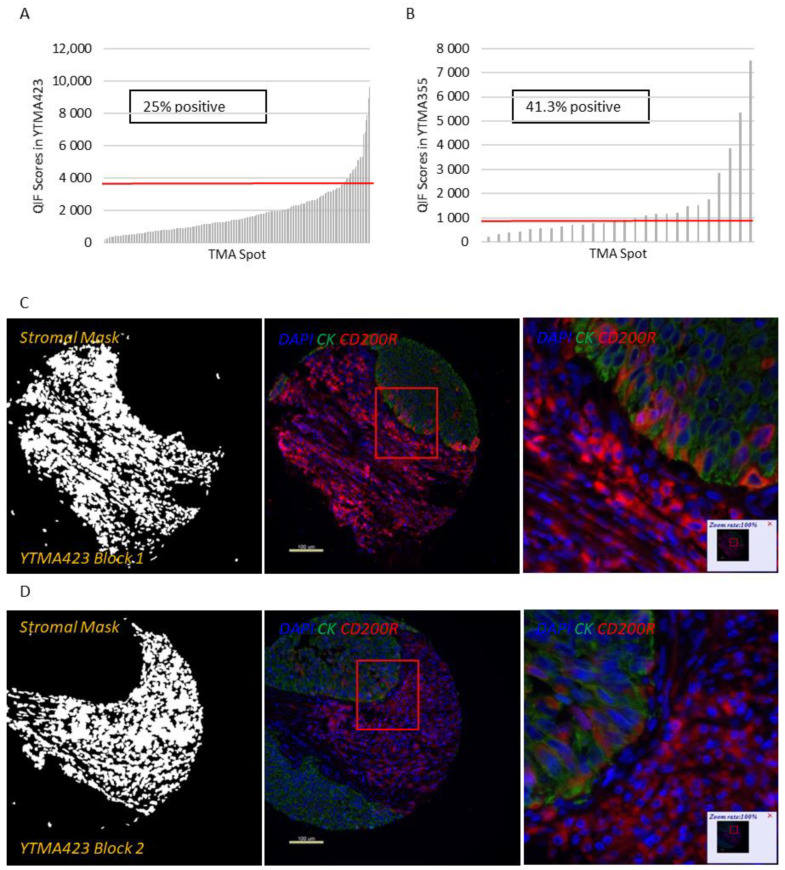
Patterns of expression and distribution of CD200R in lung cancer. (**A**,**B**) Dynamic range charts that depict CD200R expression within the stromal compartment in NSCLC (**A**) and LCNEC (**B**) patients. Red lines represent the visual cutoffs that were used to define CD200R positive cases. (**C**–**E**) Different cores from the same representative NSCLC case, stained in YTMA423 blocks 1 (**C**), 2 (**D**) and 3 (**E**). (**F**–**I**) Scatter plots indicative of intratumoral heterogeneity of CD200R in NSCLC. Scatter plots of YTMA423 block 1 versus block 2 (**F**) and block 3 (**G**). Scatter plot between serial cuts of YTMA423 block 1 (**H**). Scatter plot between tumor and stromal expression of CD200R on the same cut (**I**). Abbreviations; DAPI, 4′,6-diamino-2-phenylindole; CK, Cytokeratin.

**Figure 3 cancers-13-01024-f003:**
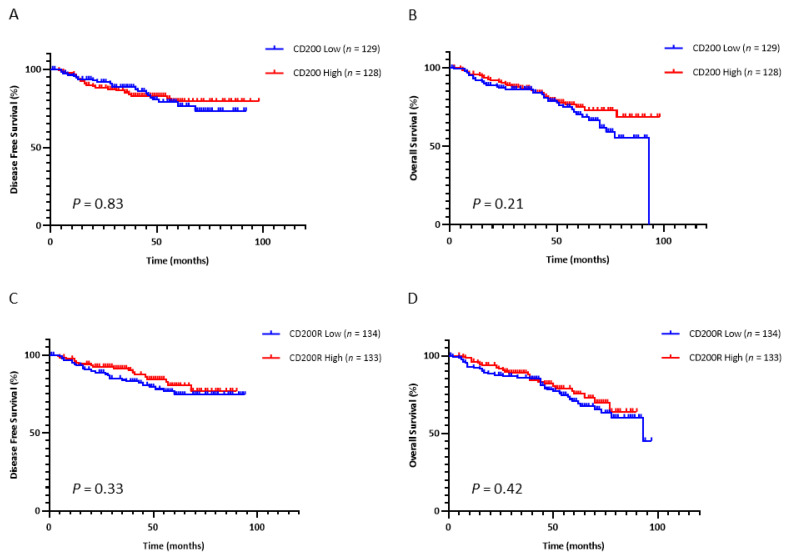
Prognostic performance of CD200 and CD200R. (**A**,**B**) Kaplan–Meier curves for disease-free (**A**) and overall survival (**B**) with respect to expression of CD200 in the tumor compartment. (**C**,**D**) Kaplan–Meier curves for disease-free (**C**) and overall survival (**D**) with respect to expression of CD200R in the stromal compartment.

**Figure 4 cancers-13-01024-f004:**
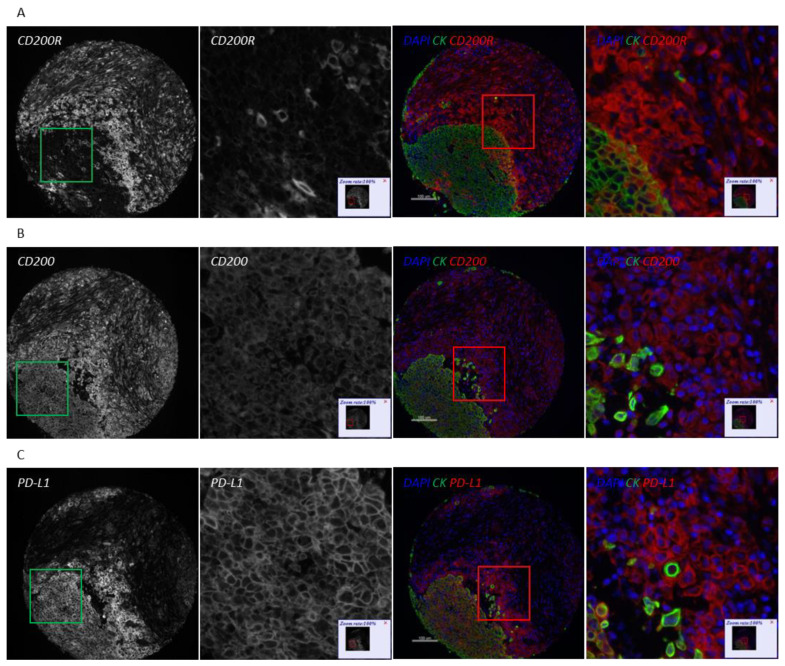
Association of CD200 and CD200R with PD-L1 on YTMA423. (**A**–**C**) Serial sections from the same NSCLC case stained for CD200R (**A**), CD200 (**B**), and PD-L1 (**C**). (**D**,**E**) Correlation between CD200 and PD-L1 measured in tumor (**D**) and stroma (**E**). (**F**–**G**) Correlation between CD200R measured in stroma and PD-L1 measured in tumor (**F**) and stroma (**G**). Abbreviations; DAPI, 4′,6-diamino-2-phenylindole; CK, Cytokeratin.

**Table 1 cancers-13-01024-t001:** Baseline characteristics of the patients in Cohort 1 (YTMA423).

Characteristic	Categories	*N* (%)	Total (%)
Age		68 (38–89)	286 (99.7)
Sex	Male	112 (39.0)	287 (100)
	Female	175 (61.0)	
Race	White	267 (93.0)	287 (100)
	Non-white	20 (7.0)	
Family cancer history	Yes	193 (67.2)	286 (99.7)
	No	93 (32.4)	
Personal cancer history	Yes	183 (63.8)	282 (98.3)
	No	99 (34.5)	
Tobacco use	Yes	247 (86.1)	287 (100)
	No	40 (13.9)	
Stage (pathological)	I	199 (69.7)	282 (98.3)
	II	66 (23.0)	
	IIIA	17 (5.9)	
Histology	Squamous	67 (23.3)	283 (98.6)
	Adenocarcinoma	209 (72.8)	
	Other	7 (2.4)	
Surgery	Yes	285 (99.3)	287 (100)
	No	2 (0.7)	

## Data Availability

The data presented in this study are available upon reasonable request from the corresponding author.

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
