# Peer review of "Quantitative Assessment of CD200 and CD200R Expression in Lung Cancer"

_cancers, 2021, doi:10.3390/cancers13051024_

Round 1

Reviewer 1 Report

Vathiotis and colleagues quantitatively assessed the expression pattern of CD200 and CD200R in lung cancers, including NSCLC and LCNEC. Approximately one third of the examined patient cases showed abundant expression for both CD200 and CD200R, which could be developed into a new clinical target in lung cancer. The information provided in the manuscript is useful for clinicians and basic research.

The correlation between CD200/CD200R and PD-1/PD-L1 is moderate/weak. It wouldn't make sense to combine these two pathways in terms of inhibitor treatment. Seems the PD-1/PD-L1 is misplaced here in this study.

What’s the criteria for the cutoff of CD200 and CD200R in figure 1 and 2, which is the foundation of this study?

The authors should spell out the abbreviations when it’s the first time in the manuscript, for example, LCNEC.

Can the authors briefly mention the current study on CD200 and CD200R in lung cancer, which will help the readers understand the significance of this study?

Author Response

Response to Reviewer 1.

Comments and Suggestions for Authors (reviewer in italics, our response in regular type)

Vathiotis and colleagues quantitatively assessed the expression pattern of CD200 and CD200R in lung cancers, including NSCLC and LCNEC. Approximately one third of the examined patient cases showed abundant expression for both CD200 and CD200R, which could be developed into a new clinical target in lung cancer. The information provided in the manuscript is useful for clinicians and basic research.

The correlation between CD200/CD200R and PD-1/PD-L1 is moderate/weak. It wouldn't make sense to combine these two pathways in terms of inhibitor treatment. Seems the PD-1/PD-L1 is misplaced here in this study.

The Authors would like to thank Reviewer 1 for his/her comments. This is a fair point raised by the Reviewer. We have rephrased both the Abstract and Simple Summary to omit the claim about combining these two pathways in terms of immune checkpoint inhibition.

What’s the criteria for the cutoff of CD200 and CD200R in figure 1 and 2, which is the foundation of this study?

The cutoff point used for CD200 and CD200R tumor positivity was visual (inspection of immunofluorescence images by eye). However, the association of CD200 and CD200R with clinicopathologic characteristics, genomic features and PD-L1 expression was assessed using continuous immunofluorescence (AQUA) scores rather than the visual cutoff point. In addition, survival analysis was performed using the median cutoff point for both CD200 and CD200R; to investigate the prognostic relevance of this pathway we have also tested visual as well as X-tile optimal cutoff points, which were not significant as well, but not mentioned in the study. At this point, it is important to state that staining and target measurement was performed in three independent TMA blocks, with each block containing one nonadjacent tumor core per patient and QIF scores were averaged for each case.

The authors should spell out the abbreviations when it’s the first time in the manuscript, for example, LCNEC.

The Reviewer is right. We have fixed this. We apologize for the inconvenience.

Can the authors briefly mention the current study on CD200 and CD200R in lung cancer, which will help the readers understand the significance of this study?

We agree with the Reviewer. To help readers understand the significance of this study, we have elaborated on the Yoshimura et al study in the Discussion section.

Reviewer 2 Report

In the study entitled " Quantitative Assessment of CD200 and CD200R Expression in Lung Cancer”, the authors used quantitative immunofluorescence to characterize the CD200/CD200R immune checkpoint in lung cancer. They developed and validated two different immunofluorescence panels with primary antibodies to detect either CD200 or CD200R on the same tissue section. In addition, even if both biomarkers were not prognostic for disease-free or overall survival in NSCLC, they showed that CD200/CD200R pathway is frequently expressed in lung cancer patients. Therefore, they suggest a potential role for targeting this pathway alone or in combination with PD-1/PD-L1 axis inhibitors.

Although the topic is very interesting and the study may have relevant implications in NSCLC treatment, there are some points that need to be addressed.

Minor Points

  • In Fig.3 perhaps there are some mistakes in the labelling of the panels. In the text, the Authors reported “Tumor CD200 expression did not significantly predict disease-free survival (DFS) (p =0.84) after surgery (Figure 3A) or overall survival (OS) (p = 0.55) (Figure 3C) in early-stage NSCLC patient”: However, in the caption they reported “(A)-(B) Kaplan-Meier curves for disease-free and overall survival with respect to expression of CD200 in the tumor compartment”. It is not clear if A and B or C are referred to CD200. The same observation is for CD200R. Moreover, it should be useful to insert CD200 or CD200R near high/low in the legends.

  • Why did the Authors not show the co-localization of CD200/CD200R and PD-L1 on the same section instead of serial sections? It would be a more elegant approach to evaluate the presence of these three biomarkers.

Round 2

Reviewer 1 Report

The authors addressed my concerns.